# Distributed Calibration of Agent-based Models

Aditi Garg
Amazon Web Services
Boston, MA

Ayush Chopra
Massachusetts Institute of Technology
Cambridge, MA

## ABSTRACT

Agent-based models (ABMs) simulate complex systems by modeling the interactions between individual agents. Calibrating ABMs to real-world data is critical for their practical utility, but is hindered by the fact that granular data is often siloed across institutions due to privacy concerns. We propose a new protocol for distributed calibration of ABMs that allows institutions to collaborate on model calibration without sharing raw data. The protocol splits the calibration neural network (CalibNN) between the data clients and a central server. Each client generates embeddings from their local data and transmits them to the server, which merges the embeddings to calibrate the ABM. Gradients are propagated back to the clients to update their local models. On preliminary experiments simulating the COVID-19 pandemic, we find the distributed protocol achieves calibration accuracy on par with centralized calibration using pooled data. This demonstrates the potential to leverage sensitive data to improve ABMs while preserving privacy.

## KEYWORDS

Agent-based Modeling, Distributed Machine Learning, Calibration

**ACM Reference Format:**
Aditi Garg and Ayush Chopra. 2024. Distributed Calibration of Agent-based Models. In *Proceedings of Workshop on Epidemiology meets Data Mining and Knowledge discovery (epiDAMIK at KDD 2024)*. ACM, New York, NY, USA, 5 pages. https://doi.org/XXXXXXX.XXXXXXX

## 1 INTRODUCTION

Agent-based models (ABM) comprise a collection of agents that act and interact within a computation world. They enable simulation of interventions by modeling the interplay between individual behaviors and environmental dynamics and hence are valuable for addressing policy questions. ABMs have helped study a multitude of complex systems across epidemiology [7, 21, 24], economics [5, 6], and disaster response [18, 19]. During the COVID-19 pandemic, ABMs helped measure the effectiveness of lockdowns [32], evaluate immunization protocols [33] and prioritize testing schedules [25]. Their utility for practical decision making requires capturing realistic environment dynamics, integrating with real-world data streams while efficiently simulating million-size population.

Their widespread adoption has been hindered by two concerns: a) computational resources required to simulate, calibrate and analyze

an ABM; b) need for microdata to generate underlying populations. Recent advances in deep learning have addressed some of the computational challenges associated with ABMs. First, vectorized programming has helped ABM scale to million-size populations [10, 13]; and differentiable programming has enabled using gradient-based learning to calibrate structural parameters [11, 13], conduct sensitivity analysis in zero-shot using autograd [31] and compose ABMs with deep neural network to ingest heterogeneous data sources for guiding the calibration [11]. As a result, it is now possible to conduct simulation, calibration and analysis of country-scale ABMs on commodity hardware. Unfortunately, improvements in the computational efficiency of ABMs is of little value if the quality of the underlying population microdata is poor.

Conventional ABMs rely on data generated using sparse summary statistics derived from real-world observations. Privacy considerations, not data sparsity, are the cause for this limited granularity as data remains siloed across diverse institutions [9]. Consider the case of epidemic modeling, which requires leveraging multimodal sources to capture feedback loop between individual behavior and disease dynamics. This include access to demographic data from the US census to define population; mobility traces from Safegraph and Google Mobility to capture interaction patterns; response on digital surveys (such as facebook) to understand compliance and risk behavior; health data (from CDC) to capture individual susceptibility; insurance and employment data (from bureau of labor statistics) to contextualize socio-economic considerations. Assimilating these dynamic data streams into the ABM calibration process has become essential for address observational gaps and generate real-time insights.

However, such information is usually sensitive and personally identifiable, and hence securely siloed under regulatory oversight. Previous attempts to release such demographic, mobility and health data for scientific research have resulted in leaks that exposed agents' personal information [1, 14, 23]. Hence, the data is shared in form of sparse summary statistics generated using privacy methods to protect individual information on an aggregate level, as used for US census [8] and Google mobility data [3]. In practice, these privacy consideration come at utility loss which inhibits real-world utility of the released data. As ABMs continue to scale towards one-to-one representations of real-world systems, there remains a fundamental limitation in their modeling potential as long as data access is constrained. Alleviating this challenge is the focus of our research.

In this paper, we introduce a mechanism for distribution simulation that can allow the institutions to collaborate but without sharing their individual data. We leverage the composability of differentiable ABMs to calibrate simulation parameters while keeping the data decentralized. We build upon existing work in collaborative machine learning. The simulation model, initialized using synthetic population, is key on a centralized server. For each of the datasets,

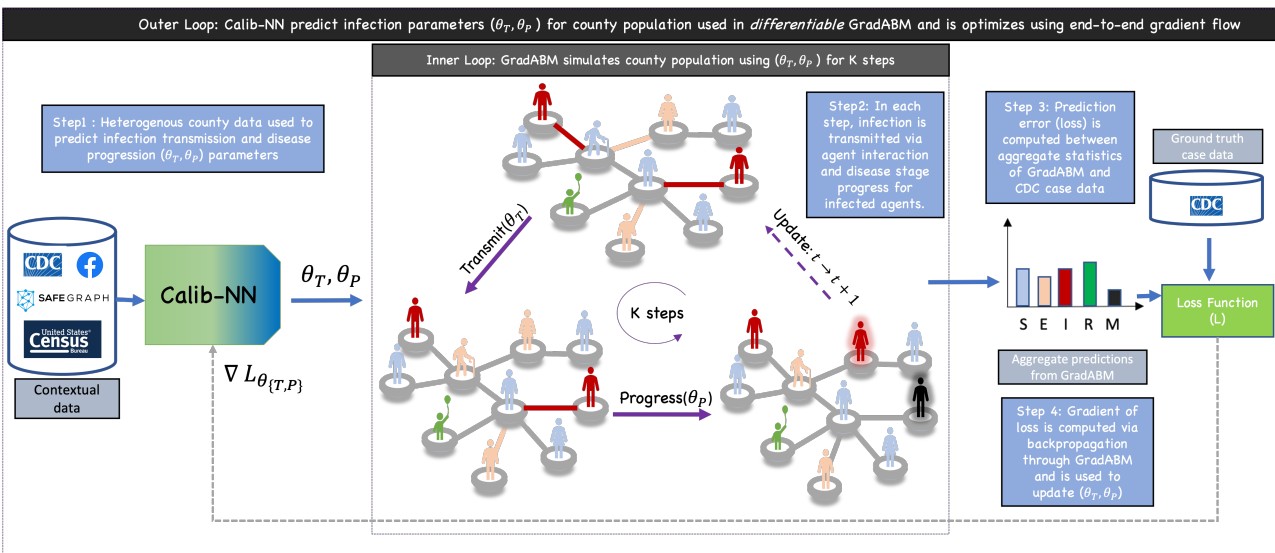

**Figure 1: Background - Centralized Calibration of ABM (source [11]). The protocol consists of four steps: i) heterogeneous macro-level population data (CDC, census, behavioral, survey) is input to a calibration model (CalibNN) to predict epidemiological parameters ($\theta_T, \theta_P$). ii): ($\theta_T, \theta_P$) are used run $K$ forward steps of the fully-differentiable epidemiological model (GradABM) which simulates micro-level infection transmission (Transmit) and disease progression (Progress) over individual contact networks. Disease statistics are aggregated (Aggregate) at end of $K$ steps to obtain the macro-level simulation output ($\hat{y}$). iii) Error between predicted $\hat{y}$ and real-world case statistics ($y$) is used to define a loss ($L(\hat{y}; \hat{y})$). iv) Gradient of this loss is computed by automatic differentiation through the micro-level GradABM to update weights of macro-level CalibNN using gradient descent. In practice, the contextual data may be siloed due privacy and logistical concerns. This work proposes a technique to execute calibration while keeping the contextual data decentralized.**

client embeddings are generated with siloed data and communicated on the server. The server aggregates the client embeddings to generate simulation parameters and execute simulation. The simulated gradients are propagated back to the clients to update CalibNN. Periodically, the CalibNN across clients are synchronized via federated averaging. This method helps achieve effective calibration, without compromising sensitivity of data. Second, we also use this method for hierarchical calibration of multiple simulations, which achieves in improved performance across different ecosystems. While our analysis is focused on ABMs, the proposed mechanism can be generalized to any differentiable simulation.

## 2 BACKGROUND

First, we review differentiable ABM and how they can be composed with deep neural networks to ingest heterogeneous data for calibration. Second, we review how this can be bridged with federated deep learning to enable calibration on siloed data.

### 2.1 Differentiable Agent-based Modeling

Consider an ABM where $X_N = F(X_0; \theta)$ where $F$ is a stochastic N-step simulator with input state $X_0$, structural parameters $\theta$ and output state $X_N$. The ABM is composed of repeated simulation steps of function $g$ s.t. $X_{t+1} = g(X_t; \theta)$. The ABM is differentiable if the gradient

$$\eta = \nabla_{\boldsymbol{\theta}} \, \mathbb{E}[F(\boldsymbol{\theta})] \tag{1}$$

exists and can be computed. In such case, for a smooth objective function $Y = c(X_n)$, the partial derivatives $dY/dX_0$ and $dY/d_\theta$ can be computed using autograd. $g$ is typically a stochastic function (eg: individual agent's probability of infection) and research in differentiable ABM has proposed methods to differentiate through discrete and continuous stochasticity [4, 11, 32] and also built frameworks built to generalize these capabilities across domains [13].

Calibration of an ABM refers to the process of estimating a set of structural parameters $\hat{\boldsymbol{\theta}}$, or a probability distribution over $\boldsymbol{\theta}$, such that $y = c(X_n)$ is consistent with real-world data. There is extensive literature on how to calibrate ABMs with techniques including approximate Bayesian computation [29], neural likelihood and posterior estimation [15], among others. When the ABM is differentiable, gradient-assisted calibration techniques can be used. This enables the ABM to be composed with neural networks into end-to-end differentiable pipelines for calibration[11] and is empirically shown to improve performance. In these protocols, the structural parameters for the ABM can be optimized by training weights of the calibration neural network. This allows to guide calibration with heterogeneous data sources; as well as jointly calibrate multiple simulators improving sample efficiency [11]. The centralized protocol is visualized in figure 1. Consider, a calibration neural network $C$ (CalibNN) with weights $\phi$ and contextual data $D$. Then, the structural parameters are generated by $\theta = C(D; \phi)$ and the ABM is executed at $F(\theta)$. The calibration gradient is:

$$\eta_c = \nabla_{\boldsymbol{\phi}} \, \mathbb{E}[F(C(D; \boldsymbol{\phi}))] \tag{2}$$

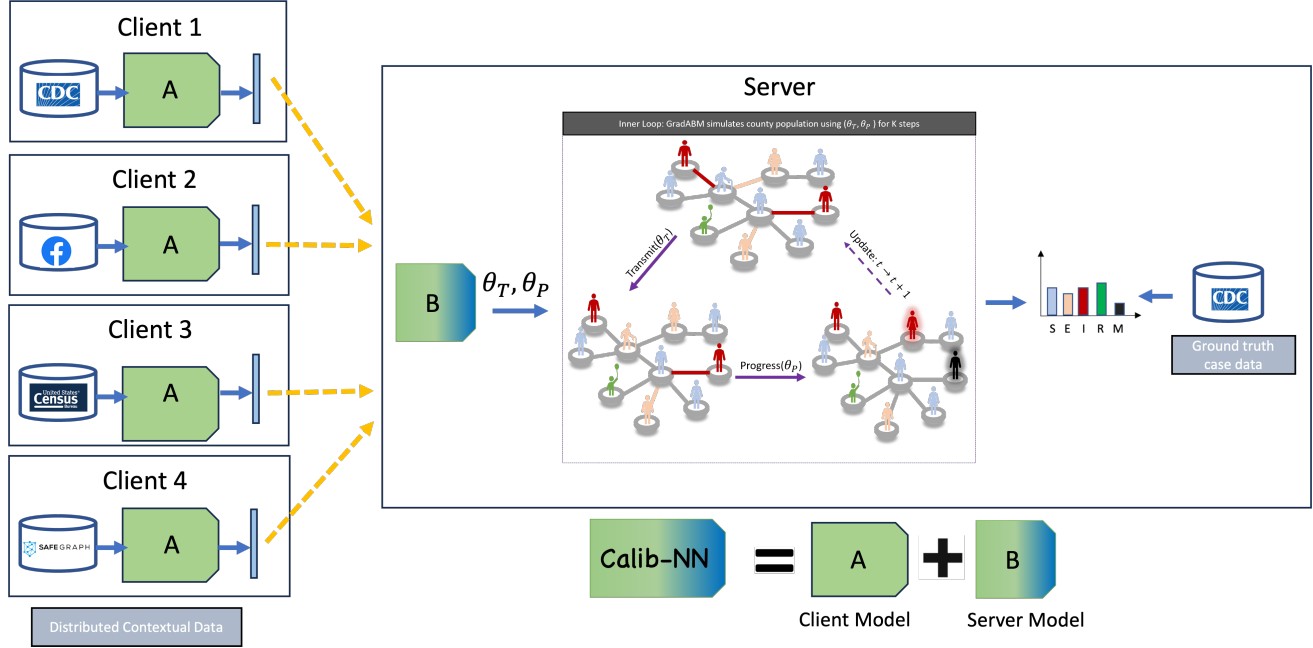

**Figure 2: Proposed - Distributed Calibration of Agent-based Models. We execute calibrate with contextual data distributed across multiple clients. To achieve this, we split CalibNN between the clients and server to ensure calibration on multi-modal data, without centralizing siloed information. Each client uses its local data to generate embeddings (in yellow) which are transmitted to server and used to predict structural parameters ($\theta_T, \theta_P$) to execute the simulation. We describe the protocol in section 4. In principle, our framework can be extended to any differentiable simulation, even beyond agent-based models.**

In practice, the contextual data $D$ can be siloed due to privacy considerations. The goal of this work is to calibrate the ABM while keeping the data decentralized. To achieve this, we draw upon work in distributed machine learning which we briefly introduce below.

## 2.2 Distributed Machine Learning

This is a setting where many clients collaborative train a model under the orchestration of a central server, while keeping the training data decentralized. As strict regulations emerge for data capture and storage, such as GDPR [17] and CCPA [2], distributed deep learning is being used to enable privacy-aware learning across a wide range of applications [16, 20, 30]. They key insight is to decentralize the model training instead of centralizing the training data. In a general distributed deep learning pipeline, there are multiple rounds of *training* and *synchronization* steps where the model is *trained* with local client data in each round and updates made by multiple clients are *synchronized* by the server into a global model. Federated learning (FL) [22, 27, 28] and split learning (SL) [12, 34] are two paradigms which provide different designs for the training and synchronization algorithms. Briefly, FL executes the training entirely on local client devices and synchronizes the models by averaging gradients on the server; while SL distributes model training between client and server and synchronizes the model by constraining clients to update shared parameters on the server model. We refer the readers to [26] for a detailed review of these paradigms. In this paper, we generalize these protocols to execute calibration of differentiable ABMs. Specifically, we extend split learning design to

distribute the CalibNN between client and server to enable learning on the siloed data during calibration.

## 3 METHOD: DISTRIBUTED AGENT-BASED MODELING

We consider a scenario where the contextual dataset $d$ is composed of multiple components $((d_1, d_2, d_3, d_4))$ which are siloed across distinct entities (called *data clients*). As motivated before, this is plausible, given the heterogeneous data requirements for ABM calibration. We consider an ABM $F$ with structural parameters $\theta$ and initial state $X_0$. The protocol is visualized in figure 2 and described below. The configuration is composed of 4 clients and 1 server. The calibration network $C$ is split as $C_a$ and $C_b$, following the split learning design [12], which are executed at client and server respectively. At the beginning, the server initializes a calibration parameters $\phi$ and transmits the model weights $\phi_a$ to all the data clients. Each episode executes as follows:

- First, each of the data client ($i = 1 to 4$) generate embeddings $E_i = C_a(d_i; \phi_a)$ and transmits the embeddings to the server.
- Second, the server merges the embeddings ($E = ([E_1, E_2, E_3, E_4])$) and generates the structural parameters $\theta = C_b(E; \phi_b)$.
- Third, the server uses $\theta$ to execute ABM simulation to generate aggregate predictions $\hat{y} = F(X_O; \theta)$ and apply a loss function $L = L(\hat{y}, y)$.
- Fourth, the gradients of the loss function $\nabla_{\phi_a} L$ are used to calibrate the structural parameters by: i) optimizing $\phi_b$ on

server using $\nabla_{\phi_b} L$ and ii) transmitting gradient ($\nabla_{\phi_b} L$) to clients to update $\phi_a$ on each client.

- Fifth, the models ($\phi_a$) at each client are periodically synchronized by the server, using federated averaging [28], to ensure consistent convergence and optimize over non-iid distributions.

We execute this process for multiple episode and the final calibrated model can be used for simulation. We empirically validate the effectiveness of the protocol in the next section.

## 4 EXPERIMENTS

To illustrate the effectiveness of our framework, we conduct experiments for COVID-19 counties of the state of Massachusetts, USA. We follow experimental protocol from [11] and consider an epidemiological ABM and calibrate R0 ($\theta_T$ in figure 1 and 2) to ground-truth data on death statistics (obtained from the CDC). The contextual data ($d$) has 4 signals:

- Mobility Signals - This record people visiting points of interest (POIs) in various regions and are obtained from Google Community Surveys
- Symptomatic surveys - This includes statistics on COVID-like illness obtained from voluntary surveys conducted by Facebook
- Symptom search data - This includes records of searches related to symptoms for multiple conditions and syndromes across the US and different states collected by Google
- Deaths and Hospitalizations - This includes data of number of hospitalizations and deaths obtained from the department of Health & Human Services and the CDC

We vertically partition the training dataset with each of the 4 signal stored in seperate clients; which is more consistent with practical considerations. In practice, the same calibration protocol can be used when the data is horizontally partitioned. The architecture of the Calibration network and the simulation design follows from [11].

We compare calibration performance with the centralized protocol (fig 1) and the proposed distributed mechanism (fig 2). We evaluate the performance using standard metrics like normal deviation(ND), root mean squared error (RMSE) and mean absolute error(MAE). Results in Table 1 shows that we distributed mechanism can preserve the performance of centralize baseline. This is an encouraging observation since it highlights the potential to use more granular signal for calibration, in the future, while ensuring data privacy.

## 5 CONCLUSION

Granular and heterogeneous data is critical for calibrating agent-based models to real-world dynamics. However, much of this data is siloed across institutions due to valid privacy concerns. We have introduced a protocol to enable distributed calibration of ABMs without centralizing raw data. By splitting the calibration network between data clients and a server, and communicating only embeddings and gradients, it allows leveraging multi-modal data while keeping it decentralized. Our experiments on a COVID-19 ABM demonstrate the distributed protocol can match the accuracy of centralized calibration using pooled data. This is a promising result,

| Calibration data | ND | RMSE | MAE |
|---|---|---|---|
| No data | 2.79 ±0.65 | 160.55 ±23.59 | 69.16 ±10.84 |
| Centralized [11] | 1.15 ±0.24 | 53.86 ±14.61 | 28.43 ±6.39 |
| Distributed (Proposed) | 1.23 ±0.35 | 54.74 ±10.61 | 30.21 ±8.45 |

**Table 1: We compare the proposed distributed calibration mechanism with two baselines: a) centralized calibration where all contextual data for calibNN is available in one-place, b) centralized calibration without using any contextual data (use gradient-based learning with CalibNN, as equation 1). We observe that the proposed method achieves consistent performance with the centralized data mechanism, while preserving privacy. We also observe that using contextual data is consistently better than not using the data. This work is promising in ability to ingest more granular data for calibration, while protecting privacy.**

as it provides a path to improve ABMs with sensitive datasets that would otherwise be inaccessible. While we focused on ABMS, the core distributed learning techniques are quite general and could be extended to other types of differentiable simulations. There are several important directions for future work. First, conducting more extensive experiments across a range of ABM applications. Second, enhancing the protocol with additional privacy safeguards such as differential privacy. Finally, exploring alternative architectures and learning algorithms to further improve efficiency and robustness. Enabling privacy-preserving use of sensitive data to enhance modeling and simulation could have a profound impact across domains from epidemiology to economics. We believe bridging techniques from multi-agent systems and collaborative learning is a fruitful direction to realize this potential. Our work provides an initial step in this direction.

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
