# OpenReview forum: "Distributed Calibration of Agent-based Models"
_KDD.org/2024/Workshop/epiDAMIK — KDD 2024 Workshop epiDAMIK_

### Official Review · Reviewer_ZBmP · 2024-06-23
**Distributed Calibration of Agent-based Models- Review**

**Rating:** 3
**Confidence:** 4

**Review:**

Summary:
The authors propose a method for distributed calibration of agent-based models (ABMs) that allows institutions to collaborate on model calibration without sharing raw data. The authors propose splitting the calibration neural network (CalibNN) between data clients and a central server. Clients generate embeddings from their local data and send them to the server, which merges them to calibrate the ABM. Gradients are sent back to clients to update local models. The method is tested on a COVID-19 ABM for Massachusetts counties, using mobility, survey, symptom search, and health data distributed across 4 clients. Results show the distributed method achieves calibration accuracy comparable to centralized calibration using pooled data.

Strong Points:
1. Via its initial results, the paper shows that the proposed distributed calibration method performs nearly as well as the centralized method.
2. The paper is well written, with clear directions explaining the motivations.

Weak points:
1. Although this is an initial work, there are many experiments that the authors should have done. For example, what is the impact of altering the number of clients on the resultant calibration? Why use SL and not FL? These analyses are very important as the initial results do not definitely validate the authors' hypothesis without robust analytic experiments.
2. As the authors state that the motivation of using embeddings instead of the deal data is for privacy, how do they ensure that the true data can't still be reconstructed from the embeddings themselves? I think that this is a big concern as an active area of research exists[1,2,3].

[1] Gupta S, Huang Y, Zhong Z, Gao T, Li K, Chen D. Recovering private text in federated learning of language models. Advances in neural information processing systems. 2022 Dec 6;35:8130-43.
[2] Zhu L, Liu Z, Han S. Deep leakage from gradients. Advances in neural information processing systems. 2019;32.
[3] Huang Y, Gupta S, Song Z, Li K, Arora S. Evaluating gradient inversion attacks and defenses in federated learning. Advances in Neural Information Processing Systems. 2021 Dec 6;34:7232-41.

---

### Official Review · Reviewer_p12Q · 2024-06-29
**Shows that ABM calibration can be done in a distributed manner.**

**Rating:** 3
**Confidence:** 4

**Review:**

**Summary**:
This paper shows that training of a differentiable model for disease spread can be done in a federated manner, with data being split across various clients. These clients send embeddings to the central server and receive gradient updates.

**Strengths:**
- The experiments are promising and show that typical regression metrics such as RMSE, MAE with the distributed model are comparable to the central model.
- The problem is well motivated: distributed training can help privacy concerns with sensitive data.

**Weaknesses:**
- The paper is not self contained and leans on [11] too often, it would be useful to describe the updates for the inner loop i.e. GradABM, is it a multi-step/single step update, how are the disease spread simulations run, gradients are calculated? The paper [11] describes 3 variations for GradABM, which one is used here is not clear.

---

### Official Review · Reviewer_Wbjm · 2024-06-30
**The distribution of calibration to preserve individual privacy**

**Rating:** 4
**Confidence:** 3

**Review:**

### Summary
This paper is concerned with the calibration of agent-based models, specifically seeking to introduce a privacy-centric method of calibration. As more thorough regulations are introduced to protect the private information of individuals, methods such as this will become increasingly important. They approach this problem by distributing a calibration neural network among the actual owners of the data, then combining the resulting parameters into the global calibration without requiring the global model to know the data itself.


### Strong Points
- The general descriptions of the method is well motivated and designed. It contributes to a pressing issue and shows promise in initial results.
- The background to distributed ML leads into the proposed method well

### Weak Points
- The descriptions of specific of the method are vague
- The experiments are limited with respect to the number of datasets are included. Is the method robust across different combinations of datasets?

### Suggestions
- Some discussion on how sensitive the method is to the number of clients would be helpful. It is quite reasonable to expect the method works for a small number of clients, but what if the model requires much more complex data? What if two clients offer similar insights, but do not offer enough data to be representative? That is, what will happen if the results of individual client calibration do not merge well?
- Do the authors expect the results to outperform centralized calibration if more sensitive data were to be used in the distributed version? This method may open the possibility to the use of more sensitive data in calibration - more sensitive than may be possible with a centralized method.

### Minor
- Fix "in in yellow" in the caption of Figure 2.
- Needs to be proofread for sentence flow errors.

---

### Official Review · Reviewer_pNpH · 2024-06-30
**Interesting idea, more details would be helpful**

**Rating:** 3
**Confidence:** 4

**Review:**

This paper proposes a new method for distributed calibration of agent-based models (ABMs), in the context of simulating the COVID-19 pandemic. The method is motivated by data privacy constraints on epidemiologically relevant data, such as mobility traces, demographics, health data, and economic data. Their method enables institutions to collaborate without sharing their data.

Their method utilizes CalibNN [11], which is prior work that enables calibration of a differentiable ABM with heterogeneous data sources. Their work differs from [11] in that they keep the datasets siloed across distinct data clients. At each stage,
1) each data client generates embeddings, based on their current copy of model weights, and transmits embeddings to the central model,
2) the model merges the embeddings to generate structural parameters,
3) uses the parameters to execute ABM simulation (disease transmission between agents) and computes the loss between aggregate predictions (COVID cases) and ground-truth,
4) the gradients of the loss are used to update the structural parameters and update the model weights on each client,
5) the model weights across clients are periodically synchronized using federated averaging.

Strengths
- The idea is interesting and may instigate fruitful conversation at the workshop
- The need for distributed calibration is well-motivated by real-world data privacy constraints
- They conduct preliminary experiments showing their distributed model achieves comparable performance to CalibNN [11] when all data is available in one place

Weaknesses
- Both their method and experiments are described briefly and would benefit from significantly more detail. For example, in their intro, they emphasize the need for more granular data - are they using individual-level data in their experiments then? How are they testing on COVID data - are they evaluating on held-out data? In their method, how are embeddings generated based on model weights? etc.
- Their method seems to rely heavily on prior work (CalibNN) and it's not clear to me how much innovation there is beyond existing methods. Based on their description, it seems like the primary innovation is that each data client has its own copy of the model weights that are periodically synced, but otherwise, embedding generation, generation of structure parameters, executing ABM, computing + applying the gradient, etc, all remain approximately the same.